environmental science, ecology

light at night, circadian rhythms, seasonal rhythms, built environment

**Author for correspondence:**
Micaela E. Martinez
e-mail: micaela.elvira.martinez@emory.edu

# Characterizing the modern light environment and its influence on circadian rhythms

Dennis Khodasevich[1], Susan Tsui[2], Darwin Keung[1], Debra J. Skene[3], Victoria Revell[4] and Micaela E. Martinez[5]

[1]Environmental Health Sciences, and [2]Department of Epidemiology, Mailman School of Public Health, Columbia University, New York, NY, USA
[3]Chronobiology, and [4]Surrey Sleep Research Centre, Faculty of Health and Medical Sciences, University of Surrey, Guildford, Surrey, UK
[5]Department of Biology, Emory University, Atlanta, GA, USA

DKh, 0000-0003-1412-8251; ST, 0000-0002-7595-2618; DKe, 0000-0003-1477-8967; DJS, 0000-0001-8202-6180; VR, 0000-0002-8809-4587; MEM, 0000-0002-9248-9450

Humans have largely supplanted natural light cycles with a variety of electric light sources and schedules misaligned with day-night cycles. Circadian disruption has been linked to a number of disease processes, but the extent of circadian disruption among the population is unknown. In this study, we measured light exposure and wrist temperature among residents of an urban area during each of the four seasons, as well as light illuminance in nearby outdoor locations. Daily light exposure was significantly lower for individuals, compared to outdoor light sensors, across all four seasons. There was also little seasonal variation in the realized photoperiod experienced by individuals, with the only significant difference occurring between winter and summer. We tested the hypothesis that differential light exposure impacts circadian phase timing, detected via the wrist temperature rhythm. To determine the influence of light exposure on circadian rhythms, we modelled the impact of morning and night-time light exposure on the timing of the maximum wrist temperature. We found that morning and night-time light exposure had significant but opposing impacts on maximum wrist temperature timing. Our results demonstrate that, within the range of exposure seen in everyday life, night-time light can delay the onset of the maximum wrist temperature, while morning light can lead to earlier onset. Our results demonstrate that humans are minimizing natural seasonal differences in light exposure, and that circadian shifts and disruptions may be a more regular occurrence in the general population than is currently recognized.

## 1. Introduction

Circadian rhythms underlie many foundational biological processes across all corners of life, ranging from prokaryotes to humans [1]. Life evolved under predictable day-night cycles, and structuring certain biological processes into 24 h cycles allowed organisms to maximize their fitness by synchronizing their internal biology with the external environment [2]. In mammals, the suprachiasmatic nucleus (SCN) serves as the central clock, receiving light information from the retina and synchronizing downstream rhythms within the organism [3]. Nearly all aspects of physiology in mammals operate under some level of circadian control, resulting in the orchestration of physiological conditions to appropriately match 24 h cycles in the environment [4]. Well-documented circadian rhythms in mammals include direct trafficking of various immune cells among the blood and organs [5], generating daily variation in gene transcription [6], controlling rhythms in the rate of protein translation [7] and altering functional responses to infection/vaccination [8,9]. Similarly, rhythms in melatonin,

DNA damage, lipid peroxidation and protein oxidation suggest there is circadian control involved in the response to oxidative stresses [10].

In addition to circadian rhythms, mammals display endogenous seasonal (i.e. circannual) rhythms in physiology and behaviour [11]. Similar to the evolution of circadian rhythms, the evolution of circannual rhythms allowed organisms to maximize their fitness by responding to predictable changes in their external environment (e.g. going into hibernation during the winter when food availability is low). Circannual rhythms in hibernating mammals are particularly apparent and well-characterized [12]. SCN neurons recognize and respond to seasonal changes in photoperiod, driving downstream circannual rhythms [13]. The molecular mechanism linking seasonal changes in day length to changes in mammalian physiology has been well-summarized [14], and great strides have been made in recent years towards further understanding the mechanism behind the mammalian circannual system [15–18]. Various studies have also revealed that aspects of human physiology display seasonal changes, including gene expression profiles in white blood cells [19], infectious disease susceptibility [20] and conception rates [21]. However, the detection of circannual rhythms in humans has been difficult and occasionally inconsistent, with the ubiquitous use of electric light in modern society being a potential explanation [22].

Although the daily and seasonal light cycles that life evolved under continue to exist, humans have largely supplanted these natural light cycles with increased time spent indoors and new light cycles built around a variety of electrical light sources. Indoor lighting places humans in an illuminance setting that would not be experienced in nature. Electrical light experienced mainly after sunset, termed light-at-night (LAN), can introduce high levels of light exposure at times that would normally be characterized by exceedingly low light exposure. Similarly, light pollution emerges from outdoor lighting and light spilling from buildings, which can result in brightness many times above moonlight intensity. Unlike other exposures, LAN does not cause direct toxicity to the body, but instead causes perturbations to the circadian and circannual systems with downstream physiological consequences [23]. Previous research has demonstrated that LAN can reduce frontal slow-wave activity during sleep [24], as well as reduced melatonin secretion and later timing of the circadian clock [25]. The evolved use of light for rhythm entrainment can have pathological consequences in the presence of artificial light, such as elevated breast cancer risk owing to light pollution and LAN [26]. A growing body of evidence suggests that chronic circadian disruption can contribute to the development of various diseases, including asthma, cancer, metabolic syndrome and cardiovascular disease [27–29].

People live with their own unique realized light cycles (RLCs), made up of a combination of natural sunlight, ambient light pollution and indoor electrical lighting. Previous observational studies have been able to characterize modern RLCs and identify effects of variations in light exposure on circadian physiology in the real world [30,31]. However, the extent to which individual variations in RLCs, both on a daily and on a seasonal scale, disrupt circadian physiology in the real world is still not well-understood. Previous research has demonstrated that daytime light exposure was lower and night-time light exposure was higher for individuals in modern constructed environments compared to natural

light-dark cycles experienced while camping [32]. Furthermore, when comparing melatonin rhythms between the summer and winter seasons, stronger seasonal differences in melatonin rhythms were found among participants in a natural lighting environment, compared to those in a modern electric lighting environment [33]. In this study, we set out to characterize the RLCs of people living within their typical modern light environment, compare these RLCs to outdoor light cycles and identify any associations between variation in light exposure and variation in circadian physiology using a non-invasive ambulatory measure of the circadian clock. Body temperature is under circadian control and has been used for many decades to monitor the circadian clock [34]. Owing to the ease of measuring body temperature using wearable devices, we used wrist temperature as a non-invasive readout of the circadian system. Wrist temperature has previously been characterized and determined to be a reliable index for evaluating circadian rhythmicity [35]. Our overall aim was to test three hypotheses. First, *people dim out their days* through time spent indoors and *light up their nights* through the use of electric light. Second, we hypothesize people experience relatively uniform light exposure throughout the year instead of the natural seasonal light cycle. Lastly, differential light exposure experienced during a normal routine can lead to shifts in circadian physiology, as detected in changes to the timing of wrist temperature rhythms.

## 2. Results

### (a) Light exposure around the clock and through the seasons

Timeseries of light illuminance measured from our outdoor sensors was highly regular, relative to individual exposure, and tightly linked to local sunrise and sunset times (figure 1*a,b*). Minimal nonzero lux readings occurred outdoors after sunset, despite the relatively high amount of light pollution expected in New York City (NYC), our primary sample site. Given the lower limit of detection of the light sensors, the illuminance of the outdoor light pollution in the study area measured less than 10 lux. Owing to the tight link with sunrise/sunset, seasonal changes in photoperiod were clearly observable from the outdoor sensors (figure 1*a*). In contrast with outdoor light, individual light cycles exhibit a high degree of variation both within and between individuals, and light exposure patterns generally did not closely align with local sunrise and sunset times (figure 1*b*). In particular, participants experienced high levels of night-time light exposure and a high degree of variability in the degree of night-time light exposure. Relative to outdoor light, individual light exposure timeseries featured a high number of days with low levels of daytime light exposure. For instance, compared to a shaded outdoor area, which regularly reached maximum daily lux values of $10^3$–$10^5$ lux, individual light exposure rarely exceeded $10^3$ lux and daily patterns were highly erratic (figure 1*b*). Lastly, many individuals exhibited low levels of light exposure throughout most of their observation weeks, with 1 or 2 days of high-intensity light exposure more closely resembling outdoor light readings, typically occurring on weekends.

Finding the average lux measurement at each time within each season revealed the general daily light exposure pattern

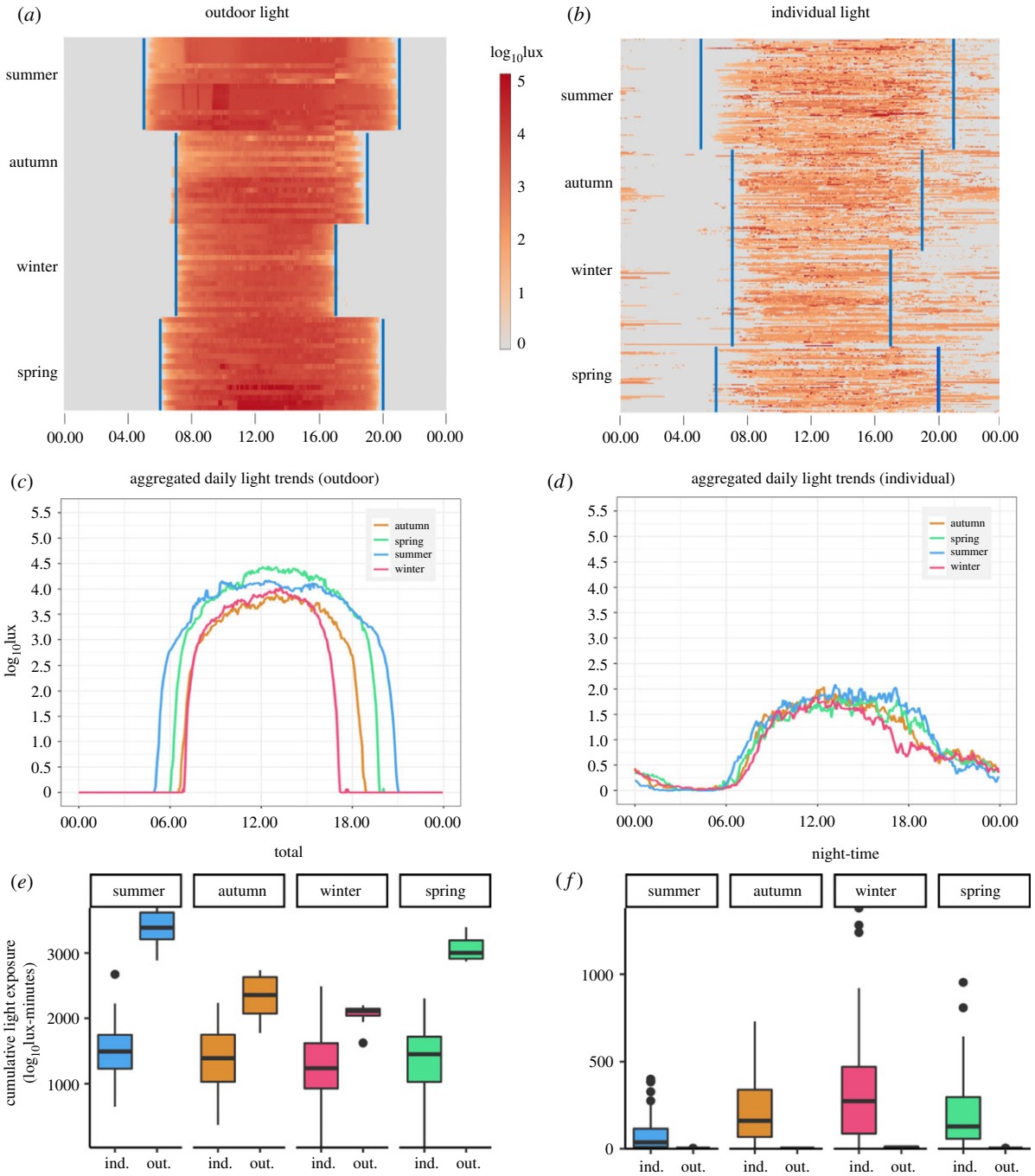

**Figure 1.** Light data characterization. Daily light exposure timeseries across all four seasons with approximate sunrise and sunset times are shown in blue from (*a*) outdoor sensors and (*b*) individual light exposure. Each row of the heatmaps contain a full 24 h period of readings from a single light sensor (single site for the outdoor data or single participant for the individual data), with rows grouped together by light sensor and by season. Individual data consists of lux readings taken at 5 min intervals over up to 7 days from study participants. Outdoor data consists of lux readings taken at 3 min intervals over 9 days from sensors located in upper Manhattan. Average light exposures at each time point for (*c*) outdoor data and (*d*) individual data. (*e*) Total daily light exposure, measured as the area-under-the-curve for the $\log_{10}$lux timeseries (in $\log_{10}$lux-minutes), comparisons of individual participant data (ind.) and outdoor data (out.) across the four seasons. (*f*) Night-time light exposure (sunset to 04.00) comparisons of individual participant data (ind.) and outdoor data (out.) across the four seasons. (Online version in colour.)

for both the outdoor environment (figure 1*c*) and individuals (figure 1*d*). While seasonal patterns are clear in the outdoor environment, these seasonal differences are diminished for the individuals. As for cumulative light exposure, individuals had relatively similar total daily light exposure from season-to-season, and the amount of light experienced was lower relative to outdoor light for all four seasons (figure 1*c*–*e*). Differences in total daily light exposure measurements (the area-under-the-curve (AUC) of the $\log_{10}$lux timeseries) are detailed in the electronic supplementary material, table S1. Total daily outdoor light exhibited a seasonal pattern with

light highest in the summer and lowest in the winter measured by outdoor sensors. However, total daily light exposure experienced by study participants exhibited no discernible seasonal pattern (figure 1*e*; electronic supplementary material, table S1). As for night-time light, most individuals experienced some night-time light, while little-to-no night-time light was detected from the outdoor sensors (figure 1*f*). Thus, we infer that night-time light exposure came from the use of indoor lighting as opposed to outdoor light pollution exposure. When we partitioned the 24 h cycle into morning, afternoon, evening and late night, we found that individuals experienced

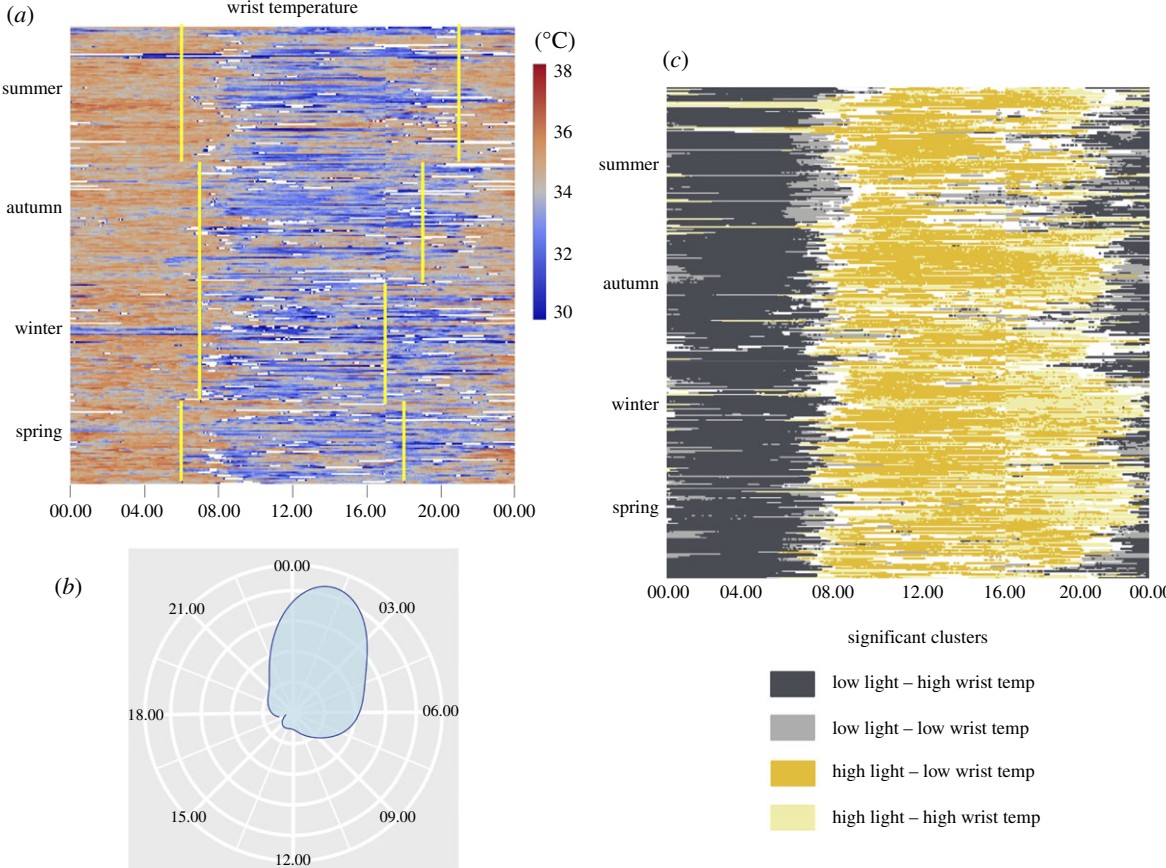

**Figure 2.** Wrist temperature characterization. (*a*) Daily wrist temperature timeseries across all four seasons with approximate sunrise and sunset times shown in yellow. Each row contains a full 24 h period of readings from a single individual, with rows grouped together by individual and by season. Individual data consists of lux readings taken at 5 min intervals over up to 7 days from study participants. Times with wrist temperatures outside of the range of normal human wrist temperature (less than 29.5°C or greater than 38.5°C) appear as white cells. (*b*) Relative frequency of daily maximum temperature timing, based on smoothed data. (*c*) Local bivariate Moran's I cluster analysis of individual light exposure and wrist temperature trend data. Significant clusters are shown in their corresponding colours, with non-significant areas shown in white. (Online version in colour.)

the most variation in light exposure late night (relative standard error (RSE) = 11.83), while the most consistent light exposure was in the afternoon (RSE = 2.06).

## (b) The effect of light-at-night and morning light on circadian physiology

Individuals exhibited a large degree of variation in daily wrist temperature but followed the same general trend of reaching a maximum wrist temperature in the late night/early morning and falling to a minimum wrist temperature in the afternoon (figure 2*a*). From this, we infer that the night-time physiological state consists of warm peripheral temperature and the daytime state consists of cool peripheral temperature. The transition from night-time to daytime physiology tended to occur in the hours around sunrise, with seasonal variation in how closely aligned the transition was to sunrise (figure 2*a*). The transition to night-time physiology was not clearly aligned with sunset and this transition time tended to be noisier among individuals and between seasons (figure 2*a*). Individual daily temperature trends were relatively noisy owing to periods with missing data and the presence of high-frequency variation in temperature within the overall 24 h trend. Daily maximum wrist temperature occurred most frequently between the hours of 00.00 and 03.00 (figure 2*b*). The cluster analysis identified significant

clusters shared among the light exposure and wrist temperature matrices (figure 2*c*). The two most frequent and biologically relevant clusters were low light/high wrist temperature clusters and high light/low wrist temperature clusters. Low light exposure and high wrist temperature was indicative of night-time physiology, while high light exposure and low wrist temperature were indicative of daytime physiology. The transition from night-time physiology to daytime physiology typically occurred between 06.00 and 08.00, while the transition from daytime physiology to night-time physiology was more variable.

The timing of the daily maximum wrist temperature was used as a biological readout of the circadian phase. We used this readout specifically because the maximum wrist temperature is typically reached during night-time hours and is less susceptible to alterations from daytime activities, compared to wrist temperature measures during the day. Our linear regression model tested the effect of night-time and morning light exposure on maximum wrist temperature timing. There was a significant effect of night-time light, which caused the maximum to occur later, as well as a significant effect of morning light, which shifted the maximum earlier (table 1). According to our best-fit linear model, the effect size of the morning and night-time light exposure were relatively similar in magnitude, suggesting that morning and night-time light may have equal but opposing effects. For example, our

**Table 1.** Linear regression model output. The dependent variable is the average timing of daily wrist temperature maximum (in decimal hours after 17.00). (Independent variables include average cumulative morning light exposure (measured in $\log_{10}$lux-minutes), average cumulative night-time light exposure (measured in $\log_{10}$lux-minutes) and season (with winter as the reference group). Residual standard error: 1.686 on 53 degrees of freedom. Multiple R-squared: 0.2769, adjusted R-squared: 0.2087. F-statistic: 4.06 on 5 and 53 d.f. p-value: 0.003414.)

|  | estimate | s.e. | t-value | Pr(>\|t\|) | |
|---|---|---|---|---|---|
| (intercept) | 10.6719 | 0.8427 | 12.664 | $<2*10^{-16}$ | *** |
| mean night-time light | 0.0056 | 0.0018 | 3.037 | 0.0037 | ** |
| mean morning light | −0.0042 | 0.0017 | −2.548 | 0.0138 | * |
| season (reference group: winter) | | | | | |
| autumn | 0.3804 | 0.6187 | 0.615 | 0.5412 | ns |
| spring | 0.3101 | 0.6570 | 0.472 | 0.6388 | ns |
| summer | 1.5371 | 0.6154 | 2.498 | 0.0156 | * |

model revealed that maximum wrist temperature timing may be shifted 1 h earlier by replacing 45 min of dim light exposure (i.e. lux < 10) in the morning with approximately 45 min of 5 $\log_{10}$lux light in the morning, which is typical of the outdoor morning. Similarly, the max timing may be shifted 1 h earlier by reducing night-time light exposure from 3 $\log_{10}$lux (i.e. typical bright indoor lighting) to < 1 $\log_{10}$lux for one hour.

Overall, more morning light and less night-time light exposure were associated with earlier maximum timing, while less morning light and more night-time light exposure were associated with later maximum timing (figure 3). There are multiple ways in which circadian rhythms can be modulated to generate the observed shift in maximum timing found by our model. One potential method is through an overall phase shift, in which the entire daily wrist temperature cycle is moved earlier or later owing to the timing of light exposure. Specifically, morning light exposure may shift the entire temperature rhythm, generating an earlier morning maximum timing (electronic supplementary material, figure S4a), while night-time light exposure shifts the maximum timing later (electronic supplementary material, figure S4b). Another potential process is through an alteration of the cycle/rhythm shape, in which the normal daily wrist temperature rhythm is temporarily distorted by light exposure. For instance, morning light exposure may lead to a faster decline in wrist temperature (electronic supplementary material, figure S4c), as opposed to a phase shift. Similarly, night-time light may lead to a delayed rise in wrist temperature and/or other distortions of the rhythm (electronic supplementary material, figure S4d).

## 3. Discussion

This study characterized daily and seasonal light exposure and wrist temperature cycles in people living within their normal environment. Individual light exposure was significantly lower than outdoor light exposure across all four seasons. Low overall light exposure probably resulted from the use of artificial light during the day and little time spent outdoors, especially on weekdays. Electric light was also used at night, resulting in higher light exposure at night, relative to outdoor conditions. Individuals exhibited a wide range of LAN exposure, ranging from undetectable to levels similar

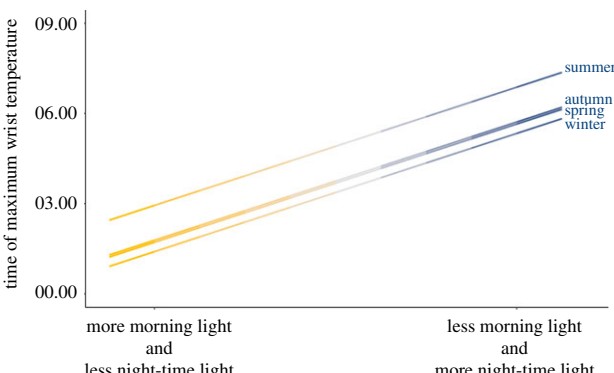

**Figure 3.** Conceptual explanation of the effect of light exposure on the timing of wrist temperature maximum. Projected change in wrist temperature maximum timing at different levels of night-time and morning light exposure, based on data outputs from the best-fit linear regression model and the range of night-time/morning light exposure estimates from the study participants. Distance between parallel lines reflects the difference in baseline wrist temperature maximum timing intercept between seasons. Horizontal axis values are based on the entire potential range of combined morning and night-time light exposure values (i.e. beginning with the combined effect of the highest morning light exposure and lowest night-time light exposure experienced by any participants, and extending to the effect of the lowest morning light exposure and highest night-time light exposure experienced by any participants). (Online version in colour.)

to that of their total daytime light exposure. Some individuals were so depauperate in daytime light and enriched in night-time light, that half of their total daily light exposure occurred at night. Owing to our sensor's inability to register light intensity values below 10 lux, we were unable to measure the effect of low-intensity LAN and outdoor light pollution. Our evaluation of light exposure, and how it is partitioned among daytime and night-time hours, supported our hypothesis that individuals living in urban environments dim out their days and light up their nights.

The results from our seasonal analysis led us to conclude that there is minimal seasonal variation in the realized photoperiod experienced by individuals. We had hypothesized that people experience uniform light exposure throughout the year. We did, however, see some variation in light exposure, with a significant difference between summer and winter. However, this summer–winter difference in realized light

exposure was one-fifth the magnitude of the same seasonal difference in the outdoor light sensors. The participants in our study lived in urban areas heavily influenced by electric light. We cannot extrapolate our seasonal results to all urban environments, because some cities may be less dependent on the electric light, facilitating more naturalistic seasonal cycles in light exposure. Furthermore, depending on economic and occupational and behavioural characteristics of a population, individuals living in other urban environments may have seasonal light exposure vastly different from that in NYC. Seasonal biology in humans is not well-understood, therefore, it is unknown what downstream effects this disconnect from natural light cycles may have on physiology and health. Our work supports previous research that has revealed a pronounced disconnect between the modern human light environment and natural light cycles, as well as muted seasonal differences in circadian timing in humans living in the modern light environment [33].

Most importantly, our study revealed that differential light exposure, within the range seen in everyday life, can lead to shifts in circadian physiology within the general population. Increased night-time light shifted the wrist temperature maximum timing later, while morning light shifted it earlier. There have been numerous laboratory experiments demonstrating that drastic changes in light exposure (i.e. mimicking night shift and/or jet lag) can lead to circadian disruption [34]. To our knowledge, ours is one of the first studies to demonstrate the effect of differential light exposure on circadian rhythms in day-to-day life across all four seasons using ambulatory monitoring devices. It is important to note that individuals in our study kept relatively typical daily schedules, similar to that of a 09.00–17.00 worker. Our results suggest, if we were to survey light exposure and circadian rhythms in a broader swath of the population, we may expect to find substantial variation in circadian entrainment owing to differential light exposure and that circadian mismatch from the natural light cycle may be a more regular occurrence than has been recognized. The minimal variation in light exposure across the year among the participants in our study may also hinder the detection of circannual rhythms among humans living in modern light environments.

The large degree of variability in light exposure among individuals living in a similar geographical area highlights the importance of personal light monitoring, as opposed to outdoor sensors and satellite data. Although LAN studies are highly represented in the chronobiology literature, we found that individuals experienced a high degree of variability in light exposure, not only at night, but across all hours of the day. This variability in light exposure may have broader implications for the generalizability of chronobiology studies conducted under strict experimental conditions. With the emerging focus on personalized medicine and the use of wearable devices to study behaviour and health, we believe that the study of light exposure and circadian rhythms in real-time opens up new opportunities for individuals to harness their clock to improve health and wellbeing.

# 4. Material and methods

## (a) Recruitment/data collection

This study was conducted under Columbia University IRB (Protocol Number AAAR7297 M00Y03). We recruited 23 adult participants for this study in summer 2018. Participants were recruited via flyers placed in Upper Manhattan, NYC and Princeton, NJ. Inclusion into the study required the participants to state that they keep a relatively consistent 8–9 h daily sleep schedule and did not identify as night owls. The majority of participants were from NYC ($n = 19$), with the mean age of participants being 32.2 years (s.d. = 8.33 years). We aimed to have a representative sample of individuals living in northern Manhattan; 70% of the participants identified as women, 22% of the participants identified as Hispanic/Latino, 22% identified as Asian, 22% identified as Black/African American, 30% identified as White and 4% identified as other.

Participants were given light illuminance sensors (HOBO® UA-002-08 Pendant Temperature/Light Data Logger) and wrist temperature sensors (iButton® temperature loggers DS1922L/DS1922T). The light sensors had a lower limit of detection of 10 lux, which limited detection of low-intensity light exposure recorded. Refer to the electronic supplementary material for a photo of the sensors. Each participant wore their sensors simultaneously for a full week during each of the four seasonal sampling sessions. The seasonal sampling sessions were held during weeks surrounding summer solstice 2018, autumn equinox 2018, winter solstice 2018 and spring equinox 2019. The loss of light/temperature sensors during observation periods and dropout between seasons lowered the effective sample size to 18 participants in the summer, 16 in the autumn, 15 in the winter and 12 in the spring.

## (b) Light exposure characterization

Light exposure was measured in 5 min intervals for each week-long seasonal sampling session, while outdoor light intensity was measured in 3 min intervals over a two-week period each season. We aligned outdoor sampling sessions to match the timing of participant sampling. Outdoor HOBO sensors were hung approximately 1.5 m above ground facing north, typically on trees (refer to the electronic supplementary material for an image of the set-up). At each outdoor sampling location, one sensor was hung in a shaded location and another was hung in a well-lit location. Light illuminance, measured in lux, was $\log_{10}$ transformed for analyses. We analysed data starting at 17.00 on the first day of sampling. Individual timeseries were categorized into observation days beginning at 17.00 and ending at 16.55 the following calendar day. Observation days were used when analysing data over 24 h periods. We created heatmaps to visualize changes in light illuminance over time, with each row containing each sequential light reading from within one observation day. Rows were organized to group together sequential observation days from the same light sensor within the same season.

We quantified light exposure as the AUC for the $\log_{10}$lux timeseries using the trapezoidal rule. All AUC measurements are expressed with the unit $\log_{10}$lux-minutes. To study the seasonal variation in light exposure, we measured: (i) total daily light exposure, (ii) night-time light exposure (i.e. sunset to 04.00) and (iii) daytime light exposure (i.e. 04.00 to approximate hour of sunset) for each observation day. We calculated this for both the participant data and the outdoor data for comparison. Tukey's honest significant difference test was used to compare means across seasons. To determine the effect of differential light exposure on circadian physiology, standardized night-time (21.00–02.00) and morning (04.00–11.59) AUC was calculated for individual participants on each observation day. This was used as inputs for the linear regression models described below. In order to quantify variability in light exposure at different times of day, the AUC was lastly calculated for four fixed-duration temporal windows: morning (05.00–11.00), afternoon (11.00–17.00), evening (17.00–23.00) and late night (23.00–05.00). The relative standard error of the AUC was calculated for each temporal window.

## (c) Temperature characterization

Wrist temperature was also measured in 5 min intervals, synchronized with the light exposure measurements. We started with 367 total observation days of data: with 109 days in the summer, 96 days in the autumn, 90 days in the winter and 72 days in the spring. Temperature readings outside of the normal biological range (less than 29.5°C or greater than 38.5°C) were replaced with n.a., as we assumed these readings occurred when participants removed their device. Only observation days with less than 28% n.a. temperature readings were kept for further analysis of wrist temperature data. This left us with 294 observation days of data: with 92 days in the summer, 77 days in the autumn, 74 days in the winter and 51 days in the spring. To visualize global patterns in the relationship between light exposure and wrist temperature, time-series matrices of light and temperature were identically gridded and treated as spatially organized grids. Using these spatially organized grids, we ran a bivariate local Moran's I using queen contiguity and eight orders of contiguity using GeoDa. The Moran's I allowed us to identify significant clusters shared among the light and temperature matrices, which were plotted in qGIS 3.10.

In order to clean data for model fitting, missing wrist temperature values were filled with predictive mean matching, using the mice R package [36]. Filled wrist temperature values were then smoothed using a smoothing spline. A comparison between the smoothed and raw wrist temperature data can be seen in the electronic supplementary material, figure S3. Using smoothed data, the timing of the wrist temperature maximum was calculated for each observation day of data. Daily light exposure and wrist temperature maximum estimates from each participant in each season were then averaged, producing a single mean estimate for each participant in each season. Final model inputs included 59 observations of mean morning light exposure, mean night-time light exposure and mean maximum wrist temperature timing split up by season, with 17 observations in the summer, 15 in the autumn, 15 in the winter and 12 in the spring. The units for light exposure measurements were $\log_{10}$lux-minutes. For ease of analysis, the unit for maximum timing was decimal observation time, which started at the beginning of each observation day (17.00), and was measured as a decimal hour after the start time (e.g. 19.45 clock time was read as $165/60 = 2.75$ h). Using these inputs, a set of linear regression models were fitted to quantify the effect of morning light exposure and night-time light exposure on maximum wrist temperature timing. Models were compared using the MuMIn R package [37], and the model with the lowest Akaike information criterion corrected for small sample size was selected as the best-fit model. Detailed comparisons between the various models can be seen in the electronic supplementary material, table S3. The final model included average maximum wrist temperature timing as the dependent variable and average morning light exposure, average night-time light exposure, and season (as a categorical variable with winter as the reference group) as independent variables.

All data analysis was done in R v. 3.6.2 [38]. Figures were generated using the R packages ggplot2 [39] and plotly [40].

Data accessibility. De-identified datasets and analysis code are available from the Dryad Digital Repository: https://doi.org/10.5061/dryad. 69p8cz90j [41].

Authors' contributions. D.Kh.: data curation, formal analysis, methodology, visualization, writing—original draft, writing—review and editing; S.T.: data curation, formal analysis, investigation, software and writing—original draft; D.Ke.: data curation, formal analysis, investigation, software and writing—original draft; D.J.S.: methodology, supervision, validation, writing—review and editing; V.R.: methodology, supervision, validation, writing—review and editing; M.E.M.: conceptualization, data curation, formal analysis, funding acquisition, investigation, methodology, project administration, resources, software, supervision, validation, writing—original draft, writing—review and editing. All authors gave final approval for publication and agreed to be held accountable for the work performed therein.

Competing interests. We declare we have no competing interests.

Funding. Our research was funded by a pilot award from the NIEHS p30 Center (grant no. ES009089). M.E.M. was also supported by the Office of the Director, National Institutes of Health of the National Institutes of Health under award no. DP5OD023100.

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
