## [Peer Review File · Proceedings of the Royal Society B: Biological Sciences]

Review History

RSPB-2021-0721.R0 (Original submission)

Review form: Reviewer 1

Recommendation

Accept with minor revision (please list in comments)

Scientific importance: Is the manuscript an original and important contribution to its field?

Good

General interest: Is the paper of sufficient general interest?

Good

Quality of the paper: Is the overall quality of the paper suitable?

Good

Is the length of the paper justified?

Yes

Should the paper be seen by a specialist statistical reviewer?

No

Do you have any concerns about statistical analyses in this paper? If so, please specify them explicitly in your report.

No

It is a condition of publication that authors make their supporting data, code and materials available - either as supplementary material or hosted in an external repository. Please rate, if applicable, the supporting data on the following criteria.

Is it accessible?

Yes

Is it clear?

Yes

Is it adequate?

Yes

Do you have any ethical concerns with this paper?

No

Comments to the Author

Here Khodasevich et al present an interesting analysis of the modern light environment on circadian rhythms in humans.

The authors monitored light exposure of a group in Manhattan across the seasons and correlated this with circadian phase as detected by wrist temperature rhythms. They show that night time indoor light in particular, but also reduced daytime exposure to outdoor light delays circadian rhythms. This is an interesting analysis, but I feel certain points need to be addressed before this paper is published.

1) In all of the heat maps shown, it is difficult to understand what each line represents, is it a single 24h period from one individual, or the cohort? Or is it an average of the entire cohort for that day?

2) whilst core body temperature is an accurate readout of circadian rhythms, wrist temperature rhythms are less well characterised. As such, it would be good to include some metrics of the quality of rhythms assessed and the confidence in the phase measurement, and how these were factored into the analysis. The authors state that where the rhythm did not follow the predicted curve, the days were excluded. Please include some data on how many days were excluded, and what proportion were included in the final analysis.

3) some important literature in this area has not been discussed, such as Stothard et al., 2017, Current Biology, Wright et al., 2013 Current Biology, Moreno et al., 2015, Sci. Rep., de la Iglesia et al., 2015, JBR, Chang et al., 2015, PNAS, Chellapa et al., 2013, J Sleep Res.

The papers all deal with the effect of artificial lights and/or seasonal light exposure on circadian rhythms, and the authors findings should be placed in the context of this previous work.

Review form: Reviewer 2

Recommendation

Major revision is needed (please make suggestions in comments)

Scientific importance: Is the manuscript an original and important contribution to its field?

Excellent

General interest: Is the paper of sufficient general interest?

Good

Quality of the paper: Is the overall quality of the paper suitable?

Good

Is the length of the paper justified?

Yes

Should the paper be seen by a specialist statistical reviewer?

No

Do you have any concerns about statistical analyses in this paper? If so, please specify them explicitly in your report.

No

It is a condition of publication that authors make their supporting data, code and materials available - either as supplementary material or hosted in an external repository. Please rate, if applicable, the supporting data on the following criteria.

Is it accessible?

Yes

Is it clear?

Yes

Is it adequate?

Yes

Do you have any ethical concerns with this paper?

No

Comments to the Author

This manuscript describes light exposure and wrist temperature patterns in humans and compares it to available sunlight measured in a near by outdoor location. The paper is well written and clear and identifies the presences and timing of artificial light in the human environment. The data sets are interesting and, although not unique, they do have contribute further to our understanding of the intensity and impact of artificial light at night on human well being.

Before possible acceptance for publication, I have suggest some moderate revisions:

Major revisions:

1) The authors specifically place their data in the context of circadian, but also circannual rhythms (L.72). However, their statement "Though its molecular architecture is unknown" seems to be under selling the great advances that have been made over the last 10-20 years in this field. Granted, this literature mainly talks about the photoperiodic-resetting and output pathway, the publications of Dardente, Hazlerigg, Loudon, Simonneaux, Lincoln, Yoshimura and many others would need to be recognized here. Because the resetting (entrainment) mechanism of the seasonal (circannual) clock in mammals is very different from that of birds (with direct hypothalamic photoreception, and no evidence for melatonin input), it would be wise if the authors omit 'bird species' from line 73 and stick to mammals. The authors should than specifically continue with the only mammalian photoperiodic system, which is relevant for this paper.

What is known here in short: the SCN entrains to sunlight and changes its neuronal activity shape with daylength (work my the Meijer lab), such that the pineal melatonin is produced at night where the duration of melatonin release in the blood is long in winter and short in summer. Melatonin is perceived by MT1 receptors in the PT (which contains the circannual clock) and induces Cry. This synchronizes the PT circadian clock to lights off, which triggers EYA3 12h after lights off. In winter, melatonin presence suppresses eya3, and no TSH is produced in the PT. In summer, there is no melatonin present 12h after lights off and Eya3 is induced more every day

due to a positive feedback loop. This leads to TSH production, which induces Dio2 (and decreases dio3) in the tanycytes in summer days. This subsequently causes hypothalamic T4 to be converted to T3, which leads eventually to GnRH neuronal activation in short day breeders and GnRH neuron deactivation in long day breeders (like humans). There are two nice little reviews of this work around by Hut (Hut et al CurrBiol 2011, and 2014), which give an easy accessible overview of the work.

Given this body of literature, it seems unfair to just talk about a 'working conceptual model'.

2) Figure 1a/b these are really nice plots, but it would help if not only an individual record is shown in b, but if the average sunlight profile over time is plotted in one line graph together with the individual profiles, averaged over all subjects. This should be done for all 4 season, to get a better overview of the quantified data and the variation around it!

3) The sine fit for the wrist temperature data is perhaps not the best way to describe the data. For core body temperature, the timing of night time minimum would do a better job, after perhaps some running average smoothing. This is because daytime temperatures are greatly influenced by various activities, which may be not related to circadian entrainment (sports, weather conditions, clothing, indoor/outdoor). The night time temperatures are usually much less noisy and measured in the stable environment of the bed. I would therefore suggest that the modelling is re-run using timing of maximal wrist temperature (after perhaps some appropriate smoothing) as a similar measure as minimal normally core body temperature minimum would classically be used as a circadian phase marker.

4) It is a really a critical omission that the authors failed to cite several seminal and pioneering studies that used a similar approach to evaluate the effects of (artificial) light on human entrainment. They should therefore discuss the work by the group of Ken Wright, who compares ALAN in groups of participants in summer and winter with and without light. Equally there is similar work from the group of Horacio de la Iglesia on the effects of ALAN. Additionally, the group of Roelof Hut (Wams et al; Woelders et al) published several papers on the effect of quantified (artificial) light intensity on human entrainment in an urban environment, while the group of Jerry Siegel published seminal work on three hunter gatherer communities without artificial light. These works should all be discussed and compared to place the current results in a wider perspective.

Minor revisions:

1) Line 82: "Electrical light, experienced mainly after sunset, termed light-at-night (LAN)". Insert "mainly"

2) Line 80-84: please simplify

Line 85: replace "the lux of moonlight" by "moonlight intensity" and rewrite to avoid the ending ", termed light pollution." - the term is OK, but the sentence is not smooth.

3) Line 87: add circannual system

4) Figure 1c/d: please provide units here on the vertical axis. It is lux values integrated over time, so it should read like lux.h. Depending on the sampling interval, the values should change appropriately. Using standard units, could make this study comparable to other studies.

5) Table 1: please indicate the dependent variable clearly in the title of the table caption as well as the unit in which the variables are measured (what does $-4 \text{ E-}3$ actually mean?). Also 10 based write powers like E-3 as 10^{-3} ; we are dealing with a written text here, not with some program like excel or so.

6) Although I am confident that the authors applied the lme package appropriately, they have failed to give a comprehensive overview of the source of variation. Of the measurements taken, I can read in fig1d what the difference in integrated light was between the seasons, but there is no

quantification or clear indication what the variation was (1) between days within a participant and within season, (2) between participants. This holds equally for the wrist temperature measurements. Please try to fix this either by a table or by an additional figure.

Decision letter (RSPB-2021-0721.R0)

14-May-2021

Dear Mr Khodasevich:

Your manuscript has now been peer reviewed and the reviews have been assessed by an Associate Editor. The reviewers' comments (not including confidential comments to the Editor) and the comments from the Associate Editor are included at the end of this email for your reference. As you will see, the reviewers and the Editors have raised some concerns with your manuscript and we would like to invite you to revise your manuscript to address them.

Research ethics:

Use of animals and field studies:

It is a condition of publication that you make available the data and research materials supporting the results in the article. Please see our Data Sharing Policies (<https://royalsociety.org/journals/authors/author-guidelines/#data>). Datasets should be deposited in an appropriate publicly available repository and details of the associated accession number, link or DOI to the datasets must be included in the Data Accessibility section of the article (<https://royalsociety.org/journals/ethics-policies/data-sharing-mining/>). Reference(s) to datasets should also be included in the reference list of the article with DOIs (where available).

If you wish to submit your data to Dryad (<http://datadryad.org/>) and have not already done so you can submit your data via this link [http://datadryad.org/submit?journalID=RSPB&manu=\(Document not available\)](http://datadryad.org/submit?journalID=RSPB&manu=(Document%20not%20available)), which will take you to your unique entry in the Dryad repository.

Please submit a copy of your revised paper within three weeks. If we do not hear from you within this time your manuscript will be rejected. If you are unable to meet this deadline please let us know as soon as possible, as we may be able to grant a short extension.

Best wishes,
Dr Robert Barton
<mailto:proceedingsb@royalsociety.org>

Associate Editor
Comments to Author:

Thank you for your manuscript, which has now been reviewed by two experts in the field of chronobiology. Both reviewers and I agree that the paper is scientifically interesting and well written. However, some moderate concerns were raised that require your attention.

1. We all agree that your work should be discussed much more within the context of previously published articles of a similar nature; indeed, many important papers have been omitted - please see both reviewer's comments for specific references that should be included in your revision.

2. I highlight the need to address the suggestion by Reviewer 2 for new analyses/figures, which would enhance the interpretation and impact of your work.

Reviewer(s)' Comments to Author:

Referee: 1

Comments to the Author(s)

Here Khodasevich et al present an interesting analysis of the modern light environment on circadian rhythms in humans.

The authors monitored light exposure of a group in Manhattan across the seasons and correlated this with circadian phase as detected by wrist temperature rhythms. They show that night time indoor light in particular, but also reduced daytime exposure to outdoor light delays circadian rhythms. This is an interesting analysis, but I feel certain points need to be addressed before this paper is published.

1) In all of the heat maps shown, it is difficult to understand what each line represents, is it a single 24h period from one individual, or the cohort? Or is it an average of the entire cohort for that day?

2) whilst core body temperature is an accurate readout of circadian rhythms, wrist temperature rhythms are less well characterised. As such, it would be good to include some metrics of the quality of rhythms assessed and the confidence in the phase measurement, and how these were factored into the analysis. The authors state that where the rhythm did not follow the predicted curve, the days were excluded. Please include some data on how many days were excluded, and what proportion were included in the final analysis.

3) some important literature in this area has not been discussed, such as Stothard et al., 2017, *Current Biology*, Wright et al., 2013 *Current Biology*, Moreno et al., 2015, *Sci. Rep.*, de la Iglesia et al., 2015, *JBR*, Chang et al., 2015, *PNAS*, Chellapa et al., 2013, *J Sleep Res*.

The papers all deal with the effect of artificial lights and/or seasonal light exposure on circadian rhythms, and the authors findings should be placed in the context of this previous work.

Referee: 2

Comments to the Author(s)

This manuscript describes light exposure and wrist temperature patterns in humans and compares it to available sunlight measured in a near by outdoor location. The paper is well written and clear and identifies the presences and timing of artificial light in the human environment. The data sets are interesting and, although not unique, they do have contribute further to our understanding of the intensity and impact of artificial light at night on human well being.

Before possible acceptance for publication, I have suggest some moderate revisions:

Major revisions:

1) The authors specifically place their data in the context of circadian, but also circannual rhythms (L.72). However, their statement "Though its molecular architecture is unknown" seems to be under selling the great advances that have been made over the last 10-20 years in this field. Granted, this literature mainly talks about the photoperiodic-resetting and output pathway, the publications of Dardente, Hazlerigg, Loudon, Simonneaux, Lincoln, Yoshimura and many others would need to be recognized here. Because the resetting (entrainment) mechanism of the seasonal (circannual) clock in mammals is very different from that of birds (with direct hypothalamic photoreception, and no evidence for melatonin input), it would be wise if the authors omit 'bird species' from line 73 and stick to mammals. The authors should than specifically continue with the only mammalian photoperiodic system, which is relevant for this paper.

What is known here in short: the SCN entrains to sunlight and changes its neuronal activity shape with daylength (work my the Meijer lab), such that the pineal melatonin is produced at night where the duration of melatonin release in the blood is long in winter and short in summer. Melatonin is perceived by MT1 receptors in the PT (which contains the circannual clock) and induces Cry. This synchronizes the PT circadian clock to lights off, which triggers EYA3 12h after lights off. In winter, melatonin presence suppresses *eya3*, and no TSH is produced in the PT. In summer, there is no melatonin present 12h after lights off and *Eya3* is induced more every day

due to a positive feedback loop. This leads to TSH production, which induces Dio2 (and decreases dio3) in the tanycytes in summer days. This subsequently causes hypothalamic T4 to be converted to T3, which leads eventually to GnRH neuronal activation in short day breeders and GnRH neuron deactivation in long day breeders (like humans). There are two nice little reviews of this work around by Hut (Hut et al CurrBiol 2011, and 2014), which give an easy accessible overview of the work.

Given this body of literature, it seems unfair to just talk about a 'working conceptual model'.

2) Figure 1a/b these are really nice plots, but it would help if not only an individual record is shown in b, but if the average sunlight profile over time is plotted in one line graph together with the individual profiles, averaged over all subjects. This should be done for all 4 season, to get a better overview of the quantified data and the variation around it!

3) The sine fit for the wrist temperature data is perhaps not the best way to describe the data. For core body temperature, the timing of night time minimum would do a better job, after perhaps some running average smoothing. This is because daytime temperatures are greatly influenced by various activities, which may be not related to circadian entrainment (sports, weather conditions, clothing, indoor/outdoor). The night time temperatures are usually much less noisy and measured in the stable environment of the bed. I would therefore suggest that the modelling is re-run using timing of maximal wrist temperature (after perhaps some appropriate smoothing) as a similar measure as minimal normally core body temperature minimum would classically be used as a circadian phase marker.

4) It is a really a critical omission that the authors failed to cite several seminal and pioneering studies that used a similar approach to evaluate the effects of (artificial) light on human entrainment. They should therefore discuss the work by the group of Ken Wright, who compares ALAN in groups of participants in summer and winter with and without light. Equally there is similar work from the group of Horacio de la Iglesia on the effects of ALAN. Additionally, the group of Roelof Hut (Wams et al; Woelders et al) published several papers on the effect of quantified (artificial) light intensity on human entrainment in an urban environment, while the group of Jerry Siegel published seminal work on three hunter gatherer communities without artificial light. These works should all be discussed and compared to place the current results in a wider perspective.

Minor revisions:

1) Line 82: "Electrical light, experienced mainly after sunset, termed light-at-night (LAN)". Insert "mainly"

2) Line 80-84: please simplify

Line 85: replace "the lux of moonlight" by "moonlight intensity" and rewrite to avoid the ending ", termed light pollution." - the term is OK, but the sentence is not smooth.

3) Line 87: add circannual system

4) Figure 1c/d: please provide units here on the vertical axis. It is lux values integrated over time, so it should read like lux.h . Depending on the sampling interval, the values should change appropriately. Using standard units, could make this study comparable to other studies.

5) Table 1: please indicate the dependent variable clearly in the title of the table caption as well as the unit in which the variables are measured (what does $-4 \text{ E-}3$ actually mean?). Also 10 based write powers like E-3 as 10^{-3} ; we are dealing with a written text here, not with some program like excel or so.

6) Although I am confident that the authors applied the lme package appropriately, they have failed to give a comprehensive overview of the source of variation. Of the measurements taken, I can read in fig1d what the difference in integrated light was between the seasons, but there is no quantification or clear indication what the variation was (1) between days within a participant

and within season, (2) between participants. This holds equally for the wrist temperature measurements. Please try to fix this either by a table or by an additional figure.

Author's Response to Decision Letter for (RSPB-2021-0721.R0)

See Appendix A.

Decision letter (RSPB-2021-0721.R1)

23-Jun-2021

Dear Mr Khodasevich

I am pleased to inform you that your manuscript entitled "Characterizing the Modern Light Environment and its Influence on Circadian Rhythms" has been accepted for publication in Proceedings B.

Data Accessibility section

Open Access

Your article has been estimated as being 8 pages long. Our Production Office will be able to confirm the exact length at proof stage.

Paper charges

All supplementary materials accompanying an accepted article will be treated as in their final form. They will be published alongside the paper on the journal website and posted on the online

figshare repository. Files on figshare will be made available approximately one week before the accompanying article so that the supplementary material can be attributed a unique DOI.

Sincerely,
Dr Robert Barton
Editor, Proceedings B
mailto: proceedingsb@royalsociety.org

Associate Editor:

Board Member: 1

Comments to Author:

The authors have revised their manuscript and adequately addressed all of the reviewer's comments/concerns. Thank you.

Board Member: 2

Comments to Author:

(There are no comments.)

Appendix A

Dear Associate Editor and Referees,

Thank you for taking the time to read our manuscript and provide such insightful comments. We have taken the time to incorporate all of your comments into our revised manuscript, as well as responding to each individual comment below in bolded text. We believe that the suggested revisions have improved the overall quality of our manuscript and it is now ready to be considered for publication in the *Proceedings of the National Academy of Sciences*.

Note: The responses here directly reference line numbers in the final revised manuscript (in the file named: "Light_exposure_circadian_clock_Proceedings_Final_Clean.docx")

Referee: 1

Comments to the Author(s)

Here Khodasevich et al present an interesting analysis of the modern light environment on circadian rhythms in humans. The authors monitored light exposure of a group in Manhattan across the seasons and correlated this with circadian phase as detected by wrist temperature rhythms. They show that night time indoor light in particular, but also reduced daytime exposure to outdoor light delays circadian rhythms. This is an interesting analysis, but I feel certain points need to be addressed before this paper is published.

1) In all of the heat maps shown, it is difficult to understand what each line represents, is it a single 24h period from one individual, or the cohort? Or is it an average of the entire cohort for that day?

Thank you for this input on our figures. Each line of the heatmaps contain a full 24-hour period of light exposure from a single individual, on one day, during the specified season. Rows of the heatmap are further organized so that consecutive observation days from the same individual within the same season are grouped together. We have added a more detailed description to the figure keys (Fig 1a-b, Fig 2a) in order to make interpretation of the heatmaps easier. Additionally, we have added two plots (Figure 1c-1d) that show average light readings at each timepoint for all four seasons.

2) Whilst core body temperature is an accurate readout of circadian rhythms, wrist temperature rhythms are less well characterized. As such, it would be good to include some metrics of the quality of rhythms assessed and the confidence in the phase measurement, and how these were factored into the analysis. The authors state that where the rhythm did not follow the predicted curve, the days were excluded. Please include some data on how many days were excluded, and what proportion were included in the final analysis.

Thank you for your suggestion. We have added more detailed information on sample sizes before and after filtering data by proportion of missing data, specifically on lines 327-349. On the advice of referee 2, we have also adjusted

our methods to utilize a slightly more stable marker of circadian phase, the wrist temperature maximum timing.

3) Some important literature in this area has not been discussed, such as Stothard et al., 2017, Current Biology, Wright et al., 2013 Current Biology, Moreno et al., 2015, Sci. Rep., de la Iglesia et al., 2015, JBR, Chang et al., 2015, PNAS, Chellapa et al., 2013, J Sleep Res. The papers all deal with the effect of artificial lights and/or seasonal light exposure on circadian rhythms, and the authors findings should be placed in the context of this previous work.

Thank you for listing these papers, which are certainly all important to the topic. We made sure to expand our introduction and discussion so that we could reference and discuss these listed papers.

Referee: 2

Comments to the Author(s)

This manuscript describes light exposure and wrist temperature patterns in humans and compares it to available sunlight measured in a near by outdoor location. The paper is well written and clear and identifies the presences and timing of artificial light in the human environment. The data sets are interesting and, although not unique, they do have contribute further to our understanding of the intensity and impact of artificial light at night on human well being. Before possible acceptance for publication, I have suggest some moderate revisions:

Major revisions:

1) The authors specifically place their data in the context of circadian, but also circannual rhythms (L.72). However, their statement "Though its molecular architecture is unknown" seems to be under selling the great advances that have been made over the last 10-20 years in this field. Granted, this literature mainly talks about the photoperiodic-resetting and output pathway, the publications of Dardente, Hazlerigg, Loudon, Simonneaux, Lincoln, Yoshimura and many others would need to be recognized here. Because the resetting (entrainment) mechanism of the seasonal (circannual) clock in mammals is very different from that of birds (with direct hypothalamic photoreception, and no evidence for melatonin input), it would be wise if the authors omit 'bird species' from line 73 and stick to mammals. The authors should than specifically continue with the only mammalian photoperiodic system, which is relevant for this paper. What is known here in short: the SCN entrains to sunlight and changes its neuronal activity shape with daylength (work my the Meijer lab), such that the pineal melatonin is produced at night where the duration of melatonin release in the blood is long in winter and short in summer. Melatonin is perceived by MT1 receptors in the PT (which contains the circannual clock) and induces Cry. This synchronizes the PT circadian clock to lights off, which triggers EYA3 12h after lights off. In winter, melatonin presence suppresses *eya3*, and no TSH is produced in the PT. In summer, there is no melatonin present 12h after lights off and *Eya3* is induced more every day due to a positive

feedback loop. This leads to TSH production, which induces Dio2 (and decreases dio3) in the tanycytes in summer days. This subsequently causes hypothalamic T4 to be converted to T3, which leads eventually to GnRH neuronal activation in short day breeders and GnRH neuron deactivation in long day breeders (like humans). There are two nice little reviews of this work around by Hut (Hut et al CurrBiol 2011, and 2014), which give an easy accessible overview of the work. Given this body of literature, it seems unfair to just talk about a 'working conceptual model'.

Thank you for this great summary of the topic. We have adjusted our introduction to include some of this information, as well as references to the noted studies, in lines 58-86 (Also copied below).

Circadian rhythms underlie many foundational biological processes across all corners of life, ranging from prokaryotes to humans [1]. Life evolved under predictable day-night cycles, and structuring certain biological processes into 24-hour cycles allowed organisms to maximize their fitness by synchronizing their internal biology with the external environment [2]. In mammals, the suprachiasmatic nucleus (SCN) serves as the central clock, receiving light information from the retina and synchronizing downstream rhythms within the organism [3]. Nearly all aspects of physiology in mammals operate under some level of circadian control, resulting in orchestration of physiological conditions to appropriately match 24-hour cycles in the environment [4]. Well-documented circadian rhythms in mammals include direct trafficking of various immune cells among the blood and organs [5], generating daily variation in gene transcription [6], controlling rhythms in the rate of protein translation [7], and altering functional responses to infection/vaccination [8,9]. Similarly, rhythms in melatonin, DNA damage, lipid peroxidation, and protein oxidation suggest there is circadian control involved in the response to oxidative stresses [10].

In addition to circadian rhythms, mammals display endogenous seasonal (i.e., circannual) rhythms in physiology and behavior [11]. Similar to the evolution of circadian rhythms, the evolution of circannual rhythms allowed organisms to maximize their fitness by responding to predictable changes in their external environment (e.g., going into hibernation during the winter when food availability is low). Circannual rhythms in hibernating mammals are particularly apparent and well-characterized [12]. SCN neurons recognize and respond to seasonal changes in photoperiod, driving downstream circannual rhythms [13]. The molecular mechanism linking seasonal changes in day length to changes in mammalian physiology has been well- summarized [14] and great strides have been made in recent years towards further understanding the mechanism behind the mammalian circannual system [15-18]. Various studies have also revealed that aspects of human physiology display seasonal changes, including gene expression profiles in white blood cells [19], infectious disease susceptibility [20], and conception rates [21]. However, the detection of circannual rhythms in humans has been difficult and occasionally inconsistent, with the ubiquitous use of electric light in modern society being a potential explanation [22].

2) Figure 1a/b these are really nice plots, but it would help if not only an individual record is

shown in b, but if the average sunlight profile over time is plotted in one line graph together with the individual profiles, averaged over all subjects. This should be done for all 4 seasons, to get a better overview of the quantified data and the variation around it!

Thank you for this input on our figures. Figure 1 has been adjusted to also include line plots showing average sunlight profile over time, for each season. Figure 1c shows average light exposure for the outdoor data by season, while Figure 1d shows average light exposure for the participants by season.

3) The sine fit for the wrist temperature data is perhaps not the best way to describe the data. For core body temperature, the timing of night time minimum would do a better job, after perhaps some running average smoothing. This is because daytime temperatures are greatly influenced by various activities, which may be not related to circadian entrainment (sports, weather conditions, clothing, indoor/outdoor). The night time temperatures are usually much less noisy and measured in the stable environment of the bed. I would therefore suggest that the modelling is re-run using timing of maximal wrist temperature (after perhaps some appropriate smoothing) as a similar measure as minimal normally core body temperature minimum would classically be used as a circadian phase marker.

Thank you for this insightful comment, which has helped us adjust our methods as described briefly here and in our manuscript. Wrist temperature was lightly smoothed using smoothing splines, and daily maximal wrist temperature timing was then calculated for each individual on each observation day. Maximal wrist temperature occurred most frequently during nighttime hours, particularly between 00:00 and 03:00, and was far less noisy of a measure of circadian phase, compared to MESOR timing. However, this lower variation in maximal temperature timing led our mixed model to have a singular fit error. In order to account for multiple observations from consecutive days from the same individual participant, we utilized an alternate methodology involving finding average light exposure estimates and maximum temperature timing for each individual and season, then using these estimated in a standard linear regression model. A more detailed description is found in the methods section of the manuscript, in lines 326-360.

4) It is a really a critical omission that the authors failed to cite several seminal and pioneering studies that used a similar approach to evaluate the effects of (artificial) light on human entrainment. They should therefore discuss the work by the group of Ken Wright, who compares ALAN in groups of participants in summer and winter with and without light. Equally there is similar work from the group of Horacio de la Iglesia on the effects of ALAN. Additionally, the group of Roelof Hut (Wams et al; Woelders et al) published several papers on the effect of quantified (artificial) light intensity on human entrainment in an urban environment, while the group of Jerry Siegel published seminal work on three hunter gatherer communities without artificial light. These works should all be discussed and compared to place the current results in a wider perspective.

Thank you, the listed papers are all important and relevant to this study. We made sure to expand our introduction and discussion sections to ensure that these studies are properly referenced.

Minor revisions:

1) Line 82: "Electrical light, experienced mainly after sunset, termed light-at-night (LAN)". Insert "mainly"

Thank you, the edit has been made and is now found on line 92.

2) Line 80-84: please simplify. Line 85: replace "the lux of moonlight" by "moonlight intensity" and rewrite to avoid the ending ", termed light pollution." - the term is OK, but the sentence is not smooth.

Thank you, this sentence has been simplified and is now located in lines 90-94.

3) Line 87: add circannual system

Thank you, this edit has been made and the sentence is now found on lines 94-96.

4) Figure 1c/d: please provide units here on the vertical axis. It is lux values integrated over time, so it should read like lux.h . Depending on the sampling interval, the values should change appropriately. Using standard units, could make this study comparable to other studies.

Thank you for noting the lack of units on figure 1c/d. The units for the y-axis are in log₁₀(lux)-minutes. This has been added and the figures are now Figures 1e/f.

5) Table 1: please indicate the dependent variable clearly in the title of the table caption as well as the unit in which the variables are measured (what does -4 E-3 actually mean??). Also 10 based write powers like E-3 as 10⁻³; we are dealing with a written text here, not with some program like excel or so.

Thank you, we have updated the table legend, as well as the methods section, to allow for easier interpretation of the table. Additionally, we have added a short example in the results section (lines 199-204) of how to interpret the results of the model.

6) Although I am confident that the authors applied the lme package appropriately, they have failed to give a comprehensive overview of the source of variation. Of the measurements taken, I can read in fig1d what the difference in integrated light was between the seasons, but there is no quantification or clear indication what the variation was (1) between days within a participant and within season, (2) between participants. This holds

equally for the wrist temperature measurements. Please try to fix this either by a table or by an additional figure.

Thank you for this comment. The light exposure boxplots (now Figures 1e-f after revisions) used data from each observation day within the specified season, so the variation is both between days within the participants within the season and between participants. For the linear regression model, the variation is between participants, as the model inputs are from average light exposure estimates/wrist temperature maximum timing from the entire observation week from each participant within each season. Additionally, we have further explained this in our methods section while describing the model inputs.